# NCM 1921, a Mixture of Several Ingredients, Including Fatty Acids and Choline, Attenuates Atopic Dermatitis in 1-Chloro-2,4-Dinitrobenzene-Treated NC/Nga Mice

**DOI:** 10.3390/nu12010165

**Published:** 2020-01-07

**Authors:** Young-Sil Lee, Won-Kyung Yang, Eun-Hee Jo, Seung Ho Shin, Young-Cheol Lee, Min-Cheol Park, Seung-Hyung Kim

**Affiliations:** 1Herbal Medicine Research Division, Korea Institute of Oriental Medicine, 1672 Yuseong-daero, Yuseong-gu, Daejeon 34054, Korea; rheeys04@gmail.com; 2Institute of Traditional Medicine & Bioscience, Daejeon University, Daejeon 34520, Korea; ywks1220@dju.kr; 3Department of Acupuncture and Moxibustion, Won-Kwang University Korean Medicine Hospital, Deokjin-gu, Jeonju-si, Jeollabuk-do 54538, Korea; freezo@wku.ac.kr; 4Sunseo Omega3, 53 Judeognong-gong-gil, Chungju-si, Chungcheongbuk-do 27462, Korea; green-grass@naver.com; 5Department of Herbology, College of Korean Medicine, Sangji University, Wonju 26339, Korea; lyc072@sangji.ac.kr; 6Department of Korean Medicine Ophthalmology and Otolaryngology and Dermatology, Won-Kwang University Korean Medicine Hospital 895 Muwang-ro, Iksan-si, Jeollabuk-do 54538, Korea

**Keywords:** atopic dermatitis, immune cells, inflammatory cytokine, skin barrier, fatty acid mixture

## Abstract

Atopic dermatitis (AD) is a chronic inflammatory skin disease in humans. In this study, we evaluated the effects of a mixture (NCM 1921) of omega-3 butter, omega-3 beef tallow oil, omega-3 lard oil, caprylic acid, lauric acid, choline, and Fe on AD in 1-chloro-2,4-dinitrobenzene (DNCB)-treated NC/Nga mice. NCM 1921 significantly ameliorated the macroscopic and microscopic signs and reduced skin thickness and mast cell incorporation in the skin lesions of mice with DNCB-induced AD. Furthermore, it reduced serum immunoglobulin E levels; reduced the number of IgE-producing B cells, peripheral blood mononuclear cells, white blood cells, and differential white blood cells; and increased the number of lymphocytes. NCM 1921 normalized the total cell number in dorsal skin tissue, the axillary lymph node, and spleen following DNCB exposure and reduced the number of CD23+/B220+ cells in the axillary lymph node and CD3+ cells in dorsal skin tissue. Moreover, it reduced the levels of interleukin (IL)-4 and IL-13 but increased the levels of interferon-γ in anti-CD3–stimulated splenocytes. Immunohistofluorescence staining showed that NCM 1921 treatment significantly increased claudin1, filaggrin, and Sirt1 protein expressions in AD skin lesions. These results suggest that NCM 1921 could be a valuable remedy for the treatment of AD.

## 1. Introduction

Atopic dermatitis (AD) is a chronic inflammatory disease of the skin. It is characterized by eczematous pruritic lesions, elevated levels of allergen-specific immunoglobulin (Ig) E antibodies in the blood, and infiltration of inflammatory cells that produce proallergic cytokines, such as interleukin (IL)-4, IL-15, and IL-13 [1,2]. The worldwide prevalence of AD is approximately 3% in infants, 10%–20% in children, and 1%–3% in adults [3]. Although genetic and environmental factors contribute to the development of AD, two hypotheses have been suggested to explain the pathogenesis of AD. The immunological hypothesis proposes that AD develops as a result of an imbalance of Th cells [1]. Th2 cells in the acute eczematic form of dermatitis cause an increase in the production of ILs, particularly IL-4, IL-5, and IL-13. This increases the level of serum IgE, which activates histamine-secreting mast cells. Another skin barrier-related hypothesis suggests that impaired functioning of the skin epidermal barrier critically influences the pathology and progression of AD [4]. Dysfunction of the skin barrier is associated with severe skin dehydration and changes in various genes/proteins related to the modulation of epidermal homeostasis, such as filaggrin and claudin 1 [5,6,7]. Epidermal water loss causes dryness and eczema, and easily can penetrate to the skin by the environmental allergens and cause itching, rashes, and later, allergic diseases [7]. Moreover, perturbed barrier function contributes toward allergic sensitization to various antigens and allergens, leading to inflammation in the skin barrier [4].

Several previous studies have reported the anti-allergic effects of oils and fatty acids from various sources. Fish oil contains omega-3 polyunsaturated fatty acids (n-3 PUFAs), such as eicosapentaenoic acid (EPA) and docosahexaenoic acid (DHA), and can reduce sensitization to allergens; alleviate the severity of AD, eczema, and asthma; and inhibit the expression of IL-1, IL-4, and IL-13 in serum [8,9,10,11]. Lauric acid and caprylic acid are components of coconut oil, which has long been used traditionally as a protective skin barrier, by an occlusive effect, and has been used to moisturize the skin and treat skin infections [12,13]. Previous studies have reported the anti-inflammatory effects of lauric acid and caprylic acid [14,15]. Choline upregulated lipid synthesis in vitro in an experimental model of transepidermal diffusion, providing a functional assessment of skin barrier properties [16]. Roth-Walter et al. reported that iron-deficient conditions can induce a Th2 environment, which contributes to the pathogenesis of allergies [17]. Based on these reports, these components are considered to play an important role in the suppression of allergic conditions. Thus, in this study, we prepared NCM 1921, a unique formula comprising omega-3 butter, omega-3 beef tallow oil, omega-3 lard oil, caprylic acid, lauric acid, choline, and Fe, which has not been tested for any ailment to date, and evaluated its therapeutic effects against AD in a murine model.

## 2. Materials and Methods

### 2.1. Sample Supplement

To prepare omega-3 butter, fat extracted from omega-3 milk (Green-grass, Chungju-si, Korea) was sterilized at 75 °C for 5 min, and then stored at 4 °C for 24 h after the addition of salt (0.5% (*w/w*)) and mixing. Omega-3 beef tallow oil and omega-3 lard oil were extracted from omega-3 beef tallow and omega-3 lard from omega-3–fed cows and pork (Green-grass, Chungju-si, Korea), respectively, by supercritical extraction. Caprylic acid was extracted from pinecones of Pinus koraiensis Siebold and Zucc (Yangpyeong, Korea) at temperatures of 65–95 °C under high pressure for 2 days by using supercritical extraction. Lauric acid from coconut oil was purchased from Chemerz Technologies Inc. (the Philippines), and choline and Fe were purchased from Kompass International SA (Beijing, China). The composition of NCM 1921, a combination of omega-3 beef tallow oil, omega-3 lard oil, caprylic acid, lauric acid, choline, and Fe, was as follows: omega-3 butter (28%, *w/w*), omega-3 beef tallow oil (42%, *w/w*), omega-3 lard oil (18%. *w/w*), caprylic acid (3.5%, *w/w*), lauric acid (6.5%, *w/w*), choline (1.5%, *w/w*), and Fe (0.5%, *w/w*). 

### 2.2. Animal Experiments

Male NC/Nga mice (7 weeks of age) were purchased from Central Lab Animal Inc. (Seoul, Korea). The study protocol was approved by Daejon University’s Animal Care and Use Committee (DJUARB2018-041), and animal studies were performed in accordance with the established institutional guidelines regarding animal care and use and the Guide for Care and Use of Laboratory Animals (National Research Council of USA, 1996). The mice were allowed to acclimatize to their environment for 1 week. After 1 week (8 weeks of age), AD was induced by topical application of 0.2% *w/v* 2,4-dinitrochlorobenzene (DNCB, Sigma Aldrich, St. Louis, MO, USA) prepared with olive oil and acetone solution (1:3) once a week. The mice with DCNB-induced AD were divided into the following three groups (*n* = 8–10): DCNB_ control (DCNB_CTL) group, DCNB_ dexamethasone 3 mg/kg treatment (DCNB_Dexa_3 mg/kg) group, and DCNB_NCM 1921 treatment (DCNB_NCM 1921_300 mg/kg) group. The NC/Nga normal control (NC/Nga_Nor) group was treated with only vehicle (3:1 mixture of acetone and olive oil) without DNCB. The NC/Nga_Nor and DCNB_CTL groups received orally, a vehicle (phosphate-buffered saline (PBS)) with the chow diet. The DCNB_Dexa_3 mg/kg group orally received commercially available dexamethasone 3 mg/kg/day with the chow diet, and the DNCB_NCM 1921 group was given NCM 1921 (equal to 300 mg/kg when calculated based on food intake) in the chow diet daily, as described above, for 5 weeks. The doses used were representative of the nontoxic and effective range of NCM 1921 based on previous reports about individual components of NCM1921 (n-3 PUFA, caprylic, lauric acid, etc.) [18,19,20]. The severity of dermatitis and the clinical index of dermatitis were assessed according to a previously reported method [21].

### 2.3. Histological Examination of AD-Like Dermal Pathology

The ears of the mice were biopsied, embedded in paraffin wax, and cut to a thickness of 4 μm. The histopathological analysis of the lesions for determining the infiltration of inflammatory cells, such as mast cells, was performed by hematoxylin and eosin (H&E) staining and toluidine blue staining.

### 2.4. Isolation of White Blood Cells from Peripheral Blood

To analyze white blood cells (i.e., neutrophils, eosinophils, basophils, and leukocytes), blood was collected from the mice by cardiac puncture. The total cell numbers were counted using a CELL-DYN^®^ 3200 analyzer (Abbott Laboratories, Santa Clara, CA, USA).

### 2.5. Isolation of Axillary Lymph Nodes, the Spleen, and Dorsal Skin Cells

Axillary lymph nodes (ALNs) and the spleen were isolated from the mice, crushed, and filtered using a 70 μm cell strainer. After centrifugation (3000× *g*, 5 min), primary spleen cells were added to ACK lysis buffer (Red Blood Cell Lysis, Thermo Scientific, Waltham, MA, USA) to remove red blood cells by incubation for 5 min; that was followed by washing twice with RPMI-1640 medium containing 10% fetal bovine serum. The isolated primary ALNs and spleen cells were resuspended in RPMI-1640 medium containing 10% fetal bovine serum and used for the experiment. Briefly, the dorsal skin was removed from the mice, minced using scalpels, and then incubated in PBS containing 1 mg/mL collagenase IV (Sigma-C5138, Sigma, St. Louis, MO, USA) at 37 °C. After incubation for 40 min, each tissue sample was vigorously pipetted up and down to further dissociate the remaining tissue clumps. The cell suspension was filtered using a 70-mm-pore nylon cell strainer (BD Falcon, Bedford, MA, USA) and then centrifuged for 20 min at 450× *g*. The cell pellet was collected, and the cells were washed twice. Total cell number was determined using a hemocytometer chamber (Thermo Fisher Scientific, Grand Island, NY, USA). The cells obtained were stained immediately with various antibodies for flow cytometry analysis.

### 2.6. Splenocyte Isolation and Culture

Spleens of mice were removed aseptically. To obtain single-cell suspensions, red blood cells were removed using red blood cell lysis buffer. The isolated splenocytes were incubated in the presence or absence of anti-CD3 antibody (0.5 μg/mL) (eBioscience, San Diego, CA, USA) for 24 h. After incubation, the supernatants were collected to determine the levels of IL-4, IL-5, IL-13, and IFN-γ.

### 2.7. Fluorescence-Activated Cell Sorting

The minced dorsal skin of mice was incubated in PBS containing 1 mg/mL collagenase IV and 2 mg/mL dispase for 40 min at 37 °C. Later, the cells were stained with anti-CD3, anti-CD4, anti-CD8, anti-CD19, anti-CD23, anti-CD69, anti-B220, anti-CCR3, and anti-CD11b antibodies (BD Biosciences Pharmingen, San Diego, CA, USA) in staining buffer (PBS containing 1% *v/v* fetal bovine serum and 0.01% *w/v* sodium azide) for 30 min on ice. The cells were then analyzed using a fluorescence-activated cell sorting analyzer with Cell-Quest software (BD Biosciences).

### 2.8. Enzyme-Linked Immunosorbent Assay

For the assessment of IgE levels in plasma, and IL-4, IL-5, IL-13, and IFN-γ levels (R&D Systems, St. Louis, MO, USA) in the supernatant of cultured splenocytes, commercially available ELISA kits were used according to the manufacturer’s protocols.

### 2.9. Statistical Analysis

Data are presented as the means ± standard errors of the means (SEMs) and are representative of three independent experiments. One-way analysis of variance (ANOVA) and Duncan’s test were applied using Prism 7.0 (GraphPad Software Inc., San Diego, CA, USA); *p* < 0.05 was considered significant.

## 3. Results

### 3.1. The Effects of NCM 1921 on Macroscopic AD Lesions

Since AD lesions cause pruritus and itching, we determined the effects of orally administered NCM 1921 (300 mg/kg) on AD skin lesions. There was no difference in body weight among the treated groups (Figure 1A). As shown in Figure 1B,C, mouse skin showed maximum skin damage as a result of AD-triggered pruritus. However, pruritic lesions ameliorated after treatment with 300 mg/kg NCM 1921; this healing effect was comparable to that achieved with Dex treatment in the positive controls. These results indicate that NCM 1921 could efficiently reverse AD skin lesions. The histology of the dermis in AD is changed by hyperkeratosis, infiltration of inflammatory and allergenic cells, and secondary bacterial infections. As shown in Figure 1D(a–d),E, NCM 1921 was able to recover dermal thickness to almost normal levels compared to that in the control group, indicating that it reduces the hyperkeratinization of the skin and improves the histological manifestation of dermal lesions of AD. Staining of the dorsal skin with toluidine blue showed a reduction in the infiltration of mast cells, the major marker for the production of IgE, in the NCM 1921-treated group (Figure 1D(e–h),F). These findings indicate that NCM 1921 has anti-allergic activity.

### 3.2. The Effects of NCM 1921 on the Plasma Levels of IgE and IgE-Producing B Cells

IgE is the body’s specific response to allergic pathogens. It plays an important role in signaling toward immunostimulatory mechanisms and an increase in the number of lymphocytes. However, an increased IgE level is also deleterious because it causes more severe pruritus. NCM 1921 efficiently and significantly decreased the levels of IgE compared with that in the control group in accordance with the reduced number of mast cells (Figure 2A). In addition, FACS analysis revealed that NCM 1921 decreased the number of IgE-producing cells and CD23^+^/B220^+^ cells in PBMCs (Figure 2B,C). These results indicate that NCM 1921-mediated suppression of IgE production may play a crucial role in the improvement of AD.

### 3.3. The Effects of NCM 1921 on Blood Cell Populations and the Absolute Numbers of Immune Cell Subtypes in ALNs, the Spleen, and Dorsal Skin

White blood cells play a major role in the body’s immune response. Their production is triggered by foreign substances, leading to the recruitment of function-specific white blood cells [22]. Thus, we determined the numbers of white blood cells in the sera of mice with AD. As shown in Table 1, the total number of white blood cells significantly reduced. The differential white blood cell count showed a remarkable suppression of the neutrophil count but a marked increase in the numbers of lymphocytes and eosinophils. Similarly, the total cell numbers in the ALNs, spleen, and dorsal skin tissue also statistically decreased by NCM 1921. Furthermore, the various types of lymphocytes are also dominant players in the immune response to allergenic bodies. In this regard, we determined the counts of immune cells subtype in the ALN, the spleen, and dorsal skin. NCM 1921 reduced the absolute numbers of CD23+/B220+ double-positive cells, which increased in the ALNs in the control group in ALN. The absolute numbers of CD3+/CD19-, CD4+/CD8-, and CD4+/CD69+ cells did not change in ALNs and the spleen. Infiltration of inflammatory cells, such as CD3+ cells, was comparatively lower in the dorsal skin in the NCM 1921 group, although CCR3+/CD11b+ cell infiltration did not suggest significant changes. These results indicate that NCM 1921 reduces inflammatory and allergic cell numbers.

### 3.4. The Effects of NCM 1921 on the Production of Th2 Cytokines (IL-4, IL-5, and IL-13) and Th1 Cytokines (IFN-γ) by Cultured Splenocytes

The spleen contains immune cells and plays an important role in the regulation of the immune system. Th1 lymphocytes secrete IFN-γ, tumor necrosis factor (TNF)-α, and IL-2. Th2 lymphocytes secrete IL-4, IL-5, IL-10, and IL-13, which contribute to the pathology of AD. We investigated whether NCM 1921 may regulate the cytokines secreted by of Th1/Th2 cells in the AD model. As shown in Figure 3, NCM 1921 decreased the IL-4 and IL-13 levels in cultured splenocytes while increasing the level of IFN-γ compared to those in the normal controls. There was no difference in the IL-5 level in the NCM 1921 treated group compared with the control group. These results indicate that NCM 1921 can improve AD by regulating Th1 and Th2 cytokines.

### 3.5. The Effects of NCM 1921 on the Expression of Claudin1, Filaggrin, and Sirt1 in Dermal Skin Tissue

Recently, it was reported that abnormal functioning of skin barriers contributes to the pathogenesis of AD. Sirt1 promotes the expression of filaggrin, a skin barrier protein which regulates skin barrier functioning through changes in skin hydration and skin pH [23,24,25]. Claudin1, present in the tight junctions of the epidermis, is an essential component for skin barrier functioning and plays a crucial role in transepidermal water loss [26,27]. In this regard, we examined the expression levels of these proteins in the dorsal skin. As shown in Figure 4, NCM 1921 restored claudin1, filaggrin, and Sirt1 protein expressions in dorsal skin tissue, indicating that NCM 1921 may ameliorate skin barrier functioning in AD.

## 4. Discussion

AD is a chronic inflammatory disease of the skin. For typical therapeutic trials to moderate AD, corticosteroids such as dexamethasone are used in the treatment of many conditions, including skin diseases and severe allergies, but, the long-term use of corticosteroids may result in side effects. For this reason, an alternative is necessary for AD patients [28]. In this study, we examined whether NCM 1921, a combination of omega-3 butter, omega-3 beef tallow oil, omega-3 lard oil, caprylic acid, lauric acid, choline, and Fe, could improve AD symptoms, and investigated the underlying mechanism of action. Although the constituents of this formula individually possess anti-inflammatory and skin barrier–protective properties [9,10,11,12,13,14,15], NCM 1921 is a unique formula and has not been investigated to date. We found that NCM 1921 showed effects on improvement of skin clinical score and on inhibition of tissue mast cells infiltration, IgE level, immune cell subtypes in various tissues, and Th2 cytokines like dexamethasone. Also, NCM 1921 regulated filaggrin, Sirt1 and claudin1 protein expression associated with skin barrier function in dorsal skin tissue, and it was more effective on expression of these proteins than dexamethasone. These findings suggest that NCM 1921 can be useful for AD patients and might be used as a corticosteroid replacement or as a supplemental agent.

AD may occur together with many diseases, such as eczema and allergic conjunctivitis; its symptoms include lichenification, pruritus, epidermal hyperplasia, and tissue remodeling [11]. In our study, we found that NCM 1921 potently suppressed visible dermal lesions of AD. H&E staining of dorsal skin patches showed NCM 1921 ameliorated the hyperkeratosis or thickening of the stratum corneum, which is the reaction of skin cells to a manual insult, such as itching.

The infiltration of white blood cells increases in AD eczematous lesions. This is particularly due to mast cell infiltration into the diseased site, which leads to increased production of IgE and subsequent pruritus and allergic manifestations. Besides mast cells, neutrophils are involved in allergic skin inflammation. Neutrophils produce leukotriene B4 (LTB4) upon skin scratching, which is essential for their accumulation in mechanically injured skin, for the recruitment of CD4+ T cells to the sites of antigen exposure, and for the development of allergic skin inflammation [29]. In our study, toluidine blue-stained skin patches showed a decrease in the number of mast cells due to NCM 1921 treatment; they also tended to decrease the member of eosinophil (CCR3+/CD11b+) in dorsal skin. In addition, NCM 1921 strongly decreased the numbers of total white blood cells and neutrophils, and increased the number of lymphocytes in PBMCs. These results indicate that NCM 1921 manifests the allergic response by reducing the numbers of inflammatory cells and allergic cells.

IgE is an important component of allergic diseases and binds to mast cells, which leads to the secretion of various allergic mediators, such as histamine and cytokines [30]. Therefore, the lower the IgE level, the lesser the allergic responses and subsequent levels of cytokines. In the present study, consistent with the reduced mast cell numbers, NCM 1921 potently ameliorated the levels of serum IgE and the numbers of IgE-producing (CD23+/B220+) cells in PBMCs and ALNs, indicating its usefulness in the treatment of AD.

AD is generally characterized by increased numbers of infiltrating T cells and activated CD4+ or CD8+ T cells in the dermis. These lymphocytes are involved in the promotion of cell-mediated and humoral immunity through the secretion of various cytokines [31]. The problem, however, occurs when cell numbers are elevated because of the continuous presence of pathogens or allergens. In such cases, an exogenous agent with anti-allergic capabilities is required to maintain the cell numbers required. Our results show that NCM 1921 lowered the levels of inflammatory cell (CD3+ cells) infiltration in the skin dorsal and tended to lower the numbers of CD4+/CD8+ cells in ALNs and the spleen. At the same time, it restored the level of INF-γ secretion by Th1 cells, as in the NC/Nga_normal group, but lowered the levels of IL-4 and IL-13 secreted by Th2 cells, which are responsible for attracting and activating eosinophils, in addition to activating mast cells to produce more IgE [32,33], as stimulated with anti-CD3 in splenocytes. Thus, NCM 1921 seems to regulate Th1/Th2 cell numbers, which may contribute to the allergic effect of NCM 1921.

Skin barrier dysfunction is considered a primary event in AD pathogenesis. Claudin 1 is a transmembrane protein present in the tight junctions of the epidermis, which is an essential component for skin barrier function, such as transepidermal water loss. According to a recent study, reduced claudin 1 protein expression contributes to impaired skin barrier function in lesioned skin of AD patients and these changes are triggered by inflammation [6,26,27]. Sirt1 has been found to promote differentiation of normal human keratinocytes [34]. In addition, it regulates skin barrier integrity and promotes the expression of filaggrin, which is an essential protein for the development of epidermal corneocytes and has an important role in modulating epidermal homeostasis in vitro and in vivo [23,35,36]. Based on these reports, it is possible that these proteins play crucial roles in barrier functions, such as barrier maintenance. In this study, NCM 1921 increased claudin 1, filaggrin and Sirt1 protein expression levels reduced by AD in dorsal skin tissue, indicating that NCM 1921 may ameliorate skin barrier functioning in AD.

## 5. Conclusions

NCM 1921 improved AD symptoms such as itching and pruritus. In addition, it reduced the numbers of mast cells, neutrophils, and IgE-producing CD3^+^/B220^+^ cells, in addition to reducing serum levels of IgE and Th2 cytokines in mice with AD. It also improved skin barrier functioning. Collectively, the findings of the present study suggest that NCM 1921 significantly ameliorates DCNB-induced AD. Therefore, NCM 1921 could be useful in treating skin disorders, including AD, which is accompanied by marked suppressive activity on all the basic components involved in allergy and AD responses.

## Figures and Tables

**Figure 1 nutrients-12-00165-f001:**
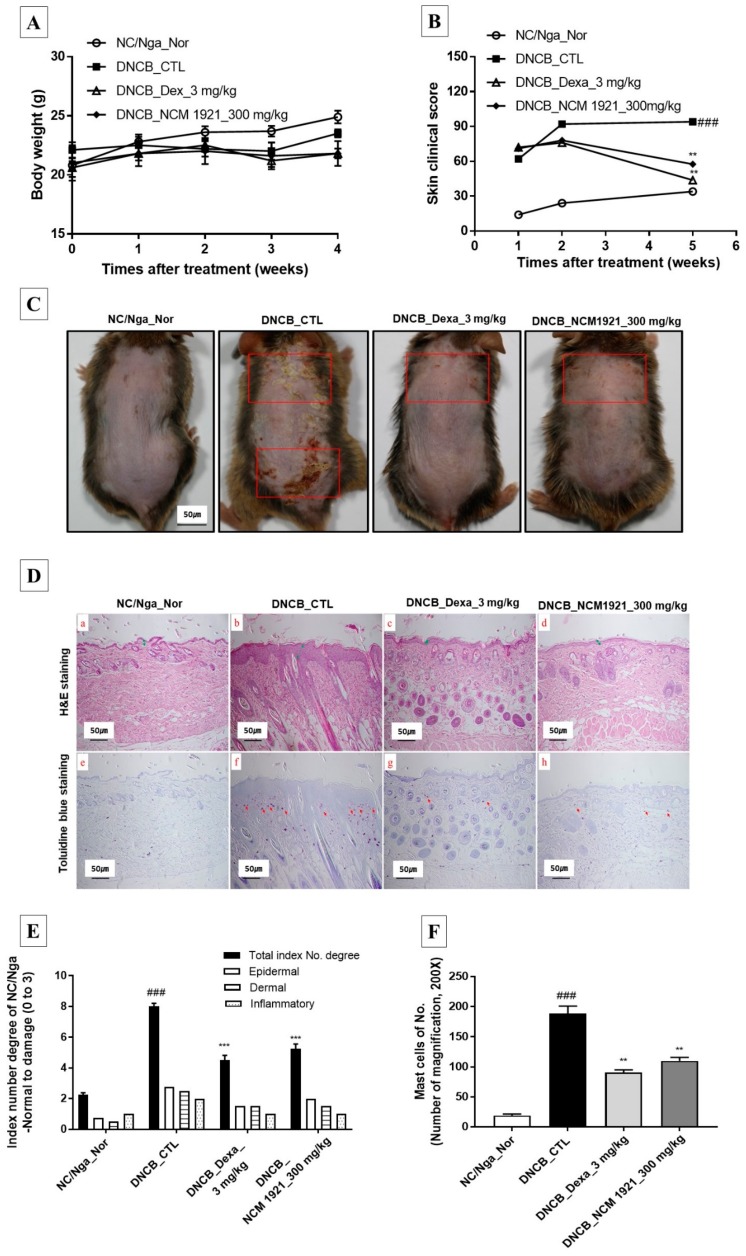
The effects of NCM 1921 on the development and the histological manifestations of atopic dermatitis in NC/Nga mice. (**A**) Body weight was measured once a week. (**B**) The severity of clinical symptoms of atopic dermatitis was evaluated macroscopically and calculated as the sum of the individual scores for the following four atopic dermatitis signs and symptoms: erythema/hemorrhage, edema, excoriation/erosion, and scaling/dryness. (**C**) Macroscopic lesions (square) in NC/Nga with atopic dermatitis induced via topical application of DNCB. (**D**) Dorsal skin sections were stained with H&E (**a**–**d**) and TB (**e**–**h**). (**E**) Total index number of degree, and (**F**) mast cell number in the dorsal skin were quantitated. H&E, hematoxylin-eosin stain; TB, toluidine blue stain; NC/Nga_Nr: normal control; DNCB_CTL: 1-chloro-2,4-dinitrobenzene (DNCB)-treated control; DNCB_Dexa 3 mg/kg: 3 mg/kg dexamethasone; DNCB_NCM 1921 300 mg/kg: 300 mg/kg NCM 1921. Values are expressed as the means ± SEMs (*n* = 6). ^###^
*p* < 0.001 compared with NC/Nga_Nor; ** *p* < 0.01 and *** *p* < 0.001 compared with DNCB_CTL as determined by ANOVA followed by multiple comparison tests.

**Figure 2 nutrients-12-00165-f002:**
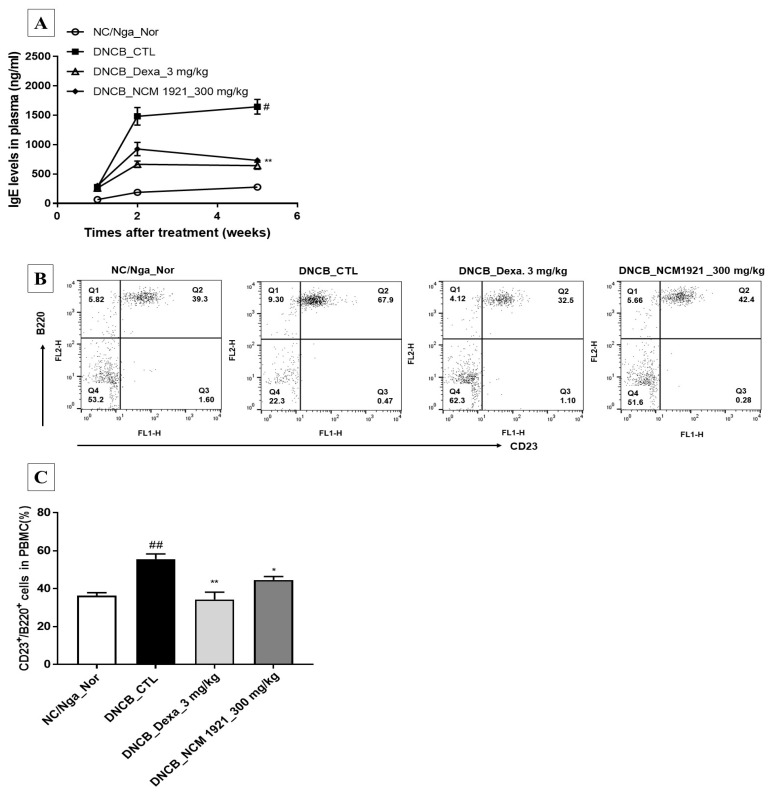
The effects of NCM 1921 on the plasma levels of immunoglobulin E (IgE) and IgE-producing B cells in NC/Nga mice with atopic dermatitis. (**A**) Total IgE levels in plasma were determined by ELISA. (**B**) The FACS analysis of dot plots indicated the percentages of B220 and CD23 double-positive cells in PBMCs. (**C**) The percentage of each cell type in CD23^+^/B220^+^ cells was calculated. Blood samples were collected from each group of mice at 11, 12, and 15 weeks of age after atopic dermatitis’s induction using DNCB and subsequent Table 1921. or dexamethasone. NC/Nga_Nor: normal control; DNCB_CTL: 1-chloro-2,4-dinitrobenzene (DNCB)-treated control; DNCB_Dexa 3 mg/kg: 3 mg/kg dexamethasone; DNCB_NCM 1921 300 mg/kg: 300 mg/kg NCM 1921. Values are expressed as the means ± SEMs (*n* = 6). ^#^
*p* < 0.05 and ^##^
*p* < 0.01 compared with NC/Nga_Nor; * *p* < 0.05 and ** *p* < 0.01 compared with DNCB_CTL as determined by ANOVA followed by multiple comparison tests.

**Figure 3 nutrients-12-00165-f003:**
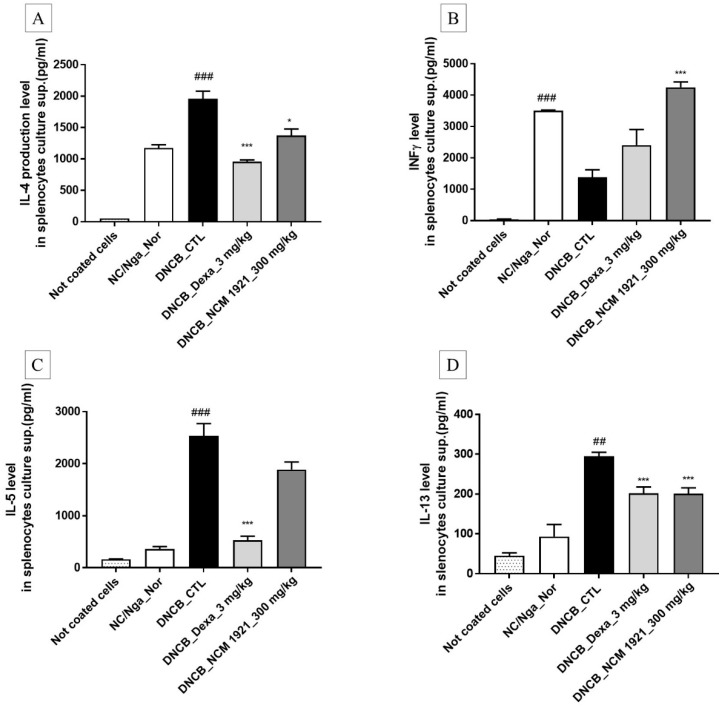
The effects of NCM 1921 on the production of Th2 and Th1 cytokines (IFN-g) by cultured splenocytes in NC/Nga mice. The levels of (**A**) IL-4, (**B**) IFN-γ, (**C**) IL-5, and (**D**) IL-13 in the cell culture supernatant were measured by ELISA. Splenocytes were cultured for 48 h at a concentration of 1 × 10^5^ cells/well by using anti-CD3 antibody–coated 96-well plates at 15 weeks of age after atopic dermatitis (AD) induction using DNCB and subsequent treatment with NCM 1921 or dexamethasone. The non-coated cells represent the negative control (i.e., no exposure to the anti-CD3 antibody). NC/Nga_Nor: normal control; DNCB_CTL: 1-chloro-2,4-dinitrobenzene (DNCB)-treated control; DNCB_Dexa 3 mg/kg: 3 mg/kg dexamethasone; DNCB_NCM 1921 300 mg/kg: 300 mg/kg NCM 1921. Values are expressed as the means ± SEMs (*n* = 6). ^##^
*p* < 0.01 and ^###^
*p* < 0.001 compared with NC/Nga_Nor; * *p* < 0.05 and *** *p* < 0.001 compared with DNCB_CTL, as determined by ANOVA followed by multiple comparison tests.

**Figure 4 nutrients-12-00165-f004:**
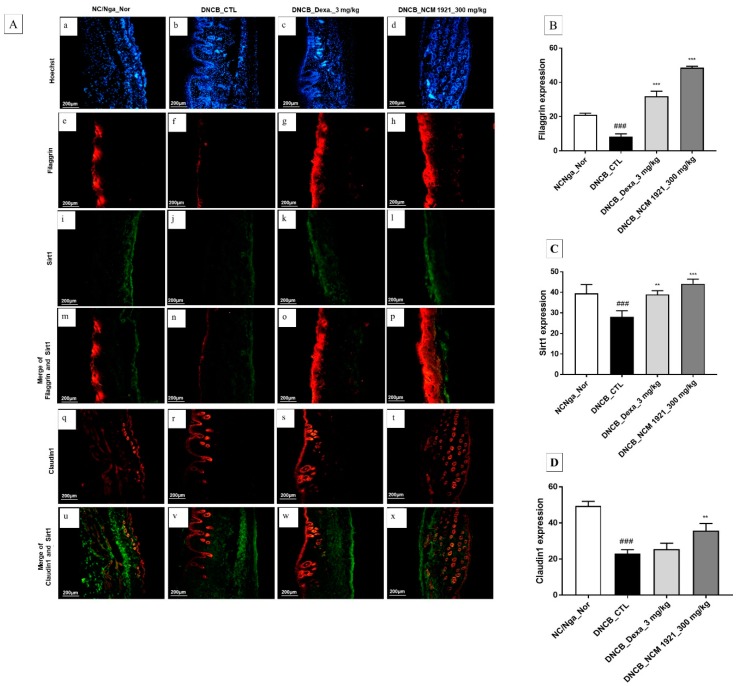
Immunohistofluorescence staining for claudin 1, filaggrin, and Sirt1 protein expression in dorsal skin tissue of NC/Nga mice. (**A**) Hoechst (blue, **a**–**d**), filaggrin (red, **e**–**h**), Sirt1 (green, **i**–**l**), Merge of filaggrin and Sirt1 (red, **m**–**p**), laudin1 (red, **q**–**t**) and Merge of claudin 1 and Sirt1 (**u**–**x**). (**a**,**e**,**i**,**m**,**q**,**u**): NC/Nga_Nor; (**b**,**f**,**j**,**n**,**r**,**v**): DNCB_CTL; (**c**,**g**,**k**,**o**,**s**,**w**): DNCB_Dexa 3 mg/kg; (**d**,**h**,**l**,**p**,**t**, **x**): DNCB_NCM 1921 300 mg/kg. Densitometric quantification of filaggrin (**B**), Sirt1 (**C**), and claudin1 (**D**) in dorsal skin tissue of animals. Fluorescence staining values quantified by image J program are represented as bars. Values are expressed as means ± SEMs (*n* = 6). ^###^
*p* < 0.001 compared with NC/Nga_Nor; ** *p* < 0.01 and *** *p* < 0.005 compared with DNCB_CTL, as determined by ANOVA followed by multiple comparison tests.

**Table 1 nutrients-12-00165-t001:** Fluorescence-activated cell sorting analysis (FACS) of immune cell subtypes in the axillary lymph nodes (ALNs), spleens, and dorsal skin tissues of DNCB-atopy dermatitis murine models.

Cell Phenotypes		DNCB			
		NC/Nga_Nor	CTL	Dexa_3 mg/kg	NCM1921_300 mg/kg
WBC	Total WBCs cells (×10^3^/µL)	3.6 ± 0.5	1.7 ± 0.1 ^##^	3.3 ± 0.5 **	3.1 ± 0.5 **
	WBCs differential counting (%)				
	Neutrophils	16.7 ± 1.4	36.7 ± 2.7	25.8 ± 0.7	25.4 ± 2.2
	Lymphocytes	79.1 ± 1.1	58.5 ± 2.3	70.3 ± 0.6	69.5 ± 2.3
	Monocytes	0.48 ± 0.1	0.43 ± 0.1	0.65 ± 0.1	0.40 ± 0.0
	Eosinophils	1.28 ± 0.3	1.17 ± 0.2	1.65 ± 0.2	2.58 ± 0.5
	Basophils	0.28 ± 0.1	0.33 ± 0.0	0.33 ± 0.0	0.25 ± 0.1
ALN	Total ALN cells (×10^4^/mL)	5.3 ± 1.6	184. ± 22.5	38.3 ± 12.2	104.3 ± 10.9
	CD3^+^/CD19^−^ (×10^5^ Cells)	0.93 ± 0.27	30.9 ± 4.57 ^###^	6.54 ± 2.11 ***	30.2 ± 5.85
	CD4^+^/CD8^+^ (×10^5^ Cells)	0.66 ± 0.13	22.2 ± 2.85 ^###^	6.28 ± 1.87 ***	19.5 ± 3.95
	CD4^+^/CD69^+^ (×10^5^ Cells)	0.59 ± 0.15	7.50 ± 1.42 ^###^	1.65 ± 0.52 **	5.84 ± 1.14
	CD23^+^/B220^+^ (×10^5^ Cells)	0.95 ± 0.24	130.2 ± 19.8 ^###^	27.0 ± 10.2 ***	62.6 ± 7.95 **
Spleen	Total spleen cells (×10^4^/mL)	665.0 ± 99.1	1585.0 ± 231.0	810.0 ± 144.0	850.0 ± 103.0
	CD3^+^/CD19^−^ (×10^5^ Cells)	265.4 ± 35.8	561.8 ± 136.2 ^#^	372.5 ± 71.7	3720 ± 41.7
	CD4^+^/CD8^+^ (× 10^5^ Cells)	119.2 ± 25.5	226.0 ± 34.5 ^#^	162.7 ± 30.7	154.9 ± 25.6
	CD4^+^/CD69^+^ (×10^5^ Cells)	7.52 ± 1.33	20.9 ± 7.3	13.7 ± 4.74	11.9 ± 1.96
	CD23^+^/B220^+^ (×10^5^ Cells)	248.5 ± 80.7	509.36 ± 110.0 ^#^	294. ± 69.3	399.4 ± 71.4
Dorsal skin	Total dorsal skin cells (×10^4^/mL)	3.5 ± 0.5	27.5 ± 2.5	9.0 ± 1.0	10.5 ± 2.5
	CD3^+^(×10^5^ Cells)	0.2 ± 0.12	3.03 ± 0.28 ^###^	1.53 ± 0.39 **	2.42 ± 0.01 *
	CCR3^+^/CD11b^+^ (×10^5^ Cells)	0.89 ± 0.06	3.79 ± 1.12 ^##^	2.07 ± 0.03	1.69 ± 0.27

Each point represents mean ± SEM values for six mice. ^#^
*p* < 0.05, ^##^
*p* < 0.01, and ^###^
*p* < 0.001 compared with NC/Nga_Nor; * *p* < 0.05, ** *p* < 0.01, and *** *p* < 0.001 compared with DNCB-CTL, as determined by one-way analysis of variance (ANOVA) followed by multiple comparison tests.

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
