# Peer review of "NCM 1921, a Mixture of Several Ingredients, Including Fatty Acids and Choline, Attenuates Atopic Dermatitis in 1-Chloro-2,4-Dinitrobenzene-Treated NC/Nga Mice"

_nutrients, 2020, doi:10.3390/nu12010165_

Round 1

Reviewer 1 Report

The manuscript submitted by Lee et al, reports attenuation of atopic dermatitis in an animal mpdel using a miwture of farry acid and cjoline. This paper is of good quality and very interesting. I only would like to point out the following;

1. Could you precise the statistical significance betweeb the two concentrations and not only vs non treated.

2. Could you precise if sahety of the mixture has been evaluated. Was efficiency evaluated for individual component of yhe mixture.

Author Response

Respone to Reviewer's was attached.

Reviewer 2 Report

Page 2, line 13: ct. Coconut oil has been used to moisturize the skin and treat skin infections [12, 13]. The reference do not seem to be appropriate.

The introduction part need to be checked giving relevant references.

Author Response

Response to reviewer's was attached.

Reviewer 3 Report

          In this manuscript, Lee et al. have demonstrated the effects of a mixture (NCM 1921) of omega-3 butter, omega-3 beef tallow oil, omega-3 lard oil, caprylic acid, lauric acid, choline, and Fe on atopic dermatitis (AD) in 1-chloro-2,4-dinitrobenzene (DNCB)-treated NC/Nga mice. The study found that NCM 1921 significantly ameliorated the macroscopic and microscopic signs, reduced skin thickness and mast cell incorporation in skin lesions, as well as reduced IgE levels, and increased the number of lymphocytes in AD mice. Moreover, it reduced the levels of IL-4 and IL-13 but increased the levels of interferon -γ in anti-CD3–stimulated splenocytes. NCM 1921 treatment also showed increased Claudin-1, Filaggrin and SIRT1 protein expression in AD skin lesions. Based on these results, the study suggests that NCM 1921 could be a valuable remedy for the treatment of AD.

            Overall, this is an interesting study. However, the major problem with the manuscript is that all the data is based on one single dose of NCM 1921. It could have been better to use a minimum two treatments for a better outcome. Additionally, there are several concerns/suggestions, which if addressed, could significantly improve the manuscript.

Is dexamethasone standard care of therapy for AD patients? If not, why authors choose to analyze the data in comparison to dexamethasone? Please provide a detailed rationale in the revised manuscript. There are no experimental details about the inclusion of NC/Nga normal mice. Please provide it. Ideally, there should be an additional group of NC/Nga normal mice treated with NCM 1921. The authors are suggested to provide complete experimental details in the revised manuscript. E.g. the rationale of treatment duration, the rationale of dose selection for NCM 1921, age of the mice, etc. The authors are suggested to add a scale bar in Fig. 1D and Fig. 4A. A superposition of fluorescence images presented in Fig. 4A could be advantageous for evaluating colocalization expression analysis for Claudin 1, Filaggrin, and Sirt1 protein. In some places, the font size does not appear to be the same. E.g. in line 249-250 “Claudin-1, present in the tight junctions of the epidermis….”. Please re-check.

Author Response

Rosponse to reviewer's comment was attached.

Round 2

Reviewer 3 Report

The authors have addressed the reviewers' concerns satisfactorily.